# Children’s Voices in Physical Activity Research: A Qualitative Review and Synthesis of UK Children’s Perspectives

**DOI:** 10.3390/ijerph19073993

**Published:** 2022-03-28

**Authors:** Lydia Emm-Collison, Rosina Cross, Maria Garcia Gonzalez, Debbie Watson, Charlie Foster, Russell Jago

**Affiliations:** 1Centre for Exercise, Nutrition & Health Sciences, School for Policy Studies, University of Bristol, Bristol BS8 1TZ, UK; maria.garcia.2018@bristol.ac.uk (M.G.G.); charlie.foster@bristol.ac.uk (C.F.); russ.jago@bristol.ac.uk (R.J.); 2Department for Health, University of Bath, Bath BA2 7AY, UK; r.cross@bath.ac.uk; 3Children and Families Research Centre, School for Policy Studies, University of Bristol, Bristol BS8 1TZ, UK; debbie.watson@bristol.ac.uk

**Keywords:** physical activity, children, qualitative systematic review

## Abstract

Background: Physical activity during childhood is associated with multiple short- and long-term health benefits. Physical activity levels decline throughout primary school emphasising a need for effective strategies to promote more activity in children. Children have rarely been involved in the intervention development process. This gap is an important omission and there is much to be learnt from existing qualitative studies with children, which could serve as a starting point for specific projects. This systematic review aimed to synthesise qualitative studies with primary school children in the United Kingdom to identify children’s perspectives on why physical activity is important, the factors that influence their physical activity and what they like when it comes to physical activity. Methods: A search of seven databases (conducted in October 2019) identified 26 papers for inclusion. Data extraction and synthesis were conducted using qualitative thematic synthesis. The quality of papers was assessed using the Critical Appraisal Skills Programme checklist for qualitative research. Results: Across the diverse range of studies, several key themes were identified in relation to the three research questions. Children have a comprehensive understanding of the various benefits of physical activity, including benefits for health, fitness and skills development. A range of social agents and practical issues influence children’s physical activity, with friend and peer influences being particularly salient. Most children like to have choice over the activities they undertake and the opportunities for creative physical play such as making up active games. Conclusions: The findings suggest that future interventions should utilize peer relationships, ensure a variety of activities are offered to cater to a broad range of children’s physical activity preferences and incorporate child-led activities where possible. The included studies also highlight a need for more diversity in qualitative research in this area, particularly in terms of ethnicity and age, and combining traditional qualitative methods with creative methods, such as photography, may provide richer insights than when using a single mode of data collection. We also highlight several methodological challenges, and in particular, the need for greater acknowledgement of the role of the researcher in qualitative work with children.

## 1. Background

Physical activity in childhood is associated with improved physical, psychological and cognitive health outcomes, including reduced adiposity and improved mental well-being [1,2]. In the United Kingdom (UK), children aged 5–18 years are recommended to engage in a minimum average of 60 min moderate-to-vigorous physical activity (MVPA) per day whilst incorporating strengthening exercise on at least 3 days per week and minimising their daily sedentary time [3]. However, many UK children do not engage in enough daily physical activity [4,5]. Longitudinal studies consistently show that MVPA declines during childhood [5,6,7], and throughout primary school MVPA declines by an average of 2.2 min per day per year [5]. 

Children’s physical activity is a complex behaviour that is influenced by a multitude of psychological (e.g., motivation), environmental (e.g., proximity to facilities) and social (e.g., the financial costs of activities) factors [8,9,10]. Over the last decade, there have been numerous efforts to minimise the decline in MVPA seen during primary school via school, family and community-based interventions, but these have had minimal effects on physical activity [11,12]. This indicates that the contexts and constructs that are central to changing behaviour in this population are not yet fully understood, and so further work is needed to identify the key factors associated with primary school children’s physical activity.

Guidance from the UK Medical Research Council emphasises the importance of supplementing the quantitative evidence base with qualitative interviews with key stakeholders to inform intervention development [13]. In the context of child physical activity, stakeholders may refer to those involved in intervention delivery (such as parents and teachers) and members of the target population (i.e., children). Children have been involved in previous qualitative studies concerning the evaluation of interventions [14], but they have rarely been engaged in formative research that contributes to intervention design. Nevertheless, there is much to be learnt from existing qualitative research with children on this topic. 

The first review of qualitative perspectives on UK physical activity was published in 2006 and included only two studies with children and adolescents [15]. In this review, it was highlighted that children aged 5–15 years were more likely to enjoy physical activity when experimentation rather than competition was encouraged, when they were given the opportunity to engage in a variety of activities, and when parents were supportive and personally enjoyed physical activity [15]. A key barrier to activity in young children was parents’ safety perceptions [15]. Whilst these issues are likely still pertinent to children’s activity engagement, there has been an abundance of qualitative studies with young children since this review was published, meaning there is much more available evidence on children’s perspectives on physical activity. Additionally, there have been a multitude of societal and cultural changes in the UK since 2006 which are likely to have impacted children’s physical activity, including changes to government funding for physical activity provision (e.g., the introduction of the school sports premium, which helps primary schools improve the quality of their provision) [16] and the technological landscape (e.g., the increased use of smartphones and social media by young people),. Recently, a qualitative synthesis of the barriers to and facilitators of children’s physical activity and sedentary behaviour was conducted in the UK [17]; however, this review focussed on younger children aged 0–6 years, and only five of the included studies include data from the children themselves. To inform the development of interventions to increase UK children’s physical activity, and given societal differences between countries globally, it is also important that this evidence is context specific. With the aim of informing future intervention development and policy for programmes focussed on the promotion and maintenance of physical activity throughout childhood, this paper presents a qualitative synthesis of UK primary school children’s perspectives on physical activity. As such, through this review we examined what is currently known about: (1) why children think physical activity is important; (2) the factors that influence children’s physical activity; and (3) what children like in terms of physical activity. 

## 2. Methods

### 2.1. Design

A systematic review of qualitative studies was conducted and reported in accordance with the Preferred Reporting Items for Systematic Reviews and Meta-Analyses (PRISMA) principles [18]. The review protocol was registered on PROSPERO (http://www.crd.york.ac.uk/PROSPERO/ registration number: CRD42019158304) (accessed on 18 November 2019). 

### 2.2. Search Strategy

An initial scoping search was carried out by the lead author (LEC) to identify existing reviews and to inform the terminology used in the final search strategy. The search strategy (Table 1) was guided by existing literature and included terms for physical activity (focus), children (population) and qualitative research (design) using boolean operators (such as OR and AND). Seven databases (CINAHL, EThOS, IBSS, Medline, PsycInfo, Scopus and Web of Science) were searched for relevant articles published between January 2004 and October 2019. An updated search was carried out on the same databases in January 2022 to identify research published between October 2019 and January 2022. The original search strategy developed in MEDLine was adapted for each database. Reference lists of relevant studies were hand searched and the other papers by authors of relevant papers were searched to identify any additional literature. 

Results were exported to Endnote X9 and screened for duplicates. Titles and abstracts were screened first, followed by the full text of the remaining papers. Papers were independently screened against the eligibility criteria by two authors (LEC and CF). As the purpose of the review is to inform the development of interventions targeting UK primary school children’s physical activity, papers were eligible for inclusion if they:reported qualitative data about physical activityinvolved children aged between 5 and11 yearsinvolved children living in the UK.

Papers were excluded if they involved participants outside of the UK due to potentially different school and cultural experiences. Studies were also excluded if they specifically recruited participants due to a medical condition or diagnosis or children living in institutional settings. This is due to the different experiences of these groups; (however, we acknowledge that the included studies may have involved some children in these situations). We also excluded qualitative data from process evaluations due to the specific focus on intervention components limiting the generalisability of the findings to the broader physical-activity context. Studies involving children outside of the age range were only included if quotes from children between aged 5 and11 years were identifiable. 

### 2.3. Quality Appraisal

The quality of studies was assessed using The Critical Appraisal Skills Programme (CASP) Qualitative Research Checklist [19]. This tool consists of 10 questions to explore whether the results of the study are valid (e.g., ‘Was the research design appropriate to meet the aims of the research?’), what the results of the study are (e.g., ‘Is there a clear statement of the findings?’) and whether the results are useful (e.g., ‘How valuable is the research?’). All included papers were assessed by the lead author (LEC) and independently assessed by a second member of the team (RC or MGG). Discrepancies were dealt with via discussion between LEC, RC and MGG. The tool was used to assess the quality of the included studies to help identify areas of methodological weakness within the current evidence base and to make recommendations for improvement in future research. Results of the quality appraisal are presented in Table 2. 

### 2.4. Data Extraction

Data were extracted using a specially created data-extraction form capturing information about the study (e.g., qualitative methodology, data analysis framework and number of interviews/focus groups) and participants (e.g., number of participants, gender, ethnicity, SES and location). This information is presented in Table 3. All primary data (participants’ quotations) from papers were extracted by LEC. Only quotes from children ages 5–11 were extracted.

### 2.5. Data Analysis

Data were synthesised and analysed in line with the principles of qualitative thematic synthesis [46]. The analysis encompassed three stages: line-by-line coding of data, the development of descriptive themes, and the generation of analytical themes (i.e., themes that go beyond those reported in the original studies). All members of the review team were involved in developing and refining the thematic framework. Data synthesis was completed using QSR NVivo (version 12).

## 3. Results

### 3.1. Study and Participant Characteristics

Through the database search, 23,628 articles were identified, with one additional article identified through searching the reference lists of relevant papers (Figure 1). The titles and abstracts of the articles were screened by LEC, and CF independently screened a random sample of 10%. The remaining full-text articles were independently screened by LEC and CF and, after discussion, 20 studies were included in this review. 

The results of this quality appraisal are presented in Table 2. All papers were deemed to make a valuable and novel contribution to the evidence base, and therefore all 26 studies were included in the review. However, only one paper clearly met all the quality indicators. The role of the researcher and their relationship with participants was one area that was consistently inadequately covered in the papers reviewed.

Table 3 provides study and participant characteristics for the 20 included studies. Of the 26 included studies, 25 were published as peer-reviewed journal articles and 1 was reported in the chapter of a doctoral thesis. All were published between 2008 and 2021. Twelve studies reported when data were collected, with data collections taking place between 2003 and 2017. Participants were from England in 17 studies, Scotland in 3 studies, and Northern Ireland in 1 study. Five studies did not report the specific region their participants were from, but four of the research groups were based in England and one was based in Wales. 

In terms of research focus, 17 studies were concerned with physical activity broadly, 1 with active play (defined as physical activity obtained through informal play) [21], 4 with physical education (PE), 3 with sport and 1 with active travel. Seven studies explored physical activity behaviours within a specific context, such as during school breaktimes or at the beach. Twelve studies used focus groups, two studies used individual interviews, two studies used both focus groups and interviews, one study used a combination of focus groups and open-ended questionnaire responses, and nine studies used visual methods (photography or drawing) in combination with focus groups. Across the studies, individual interviews lasted 15–53 min and focus groups lasted between 15 min and 1 h 38 min. Six studies did not report the duration of interviews. Almost all data were analysed using thematic, content or framework analysis. The studies involved a total of 1633 children aged 5–11 years, of which at least 608 were girls (4 studies did not report gender). Only the data from primary-school aged children (i.e., aged 5–11 years) are included in this review. Most studies did not report socio–economic status (SES) or ethnicity at the participant level, but where those were reported, they are presented in Table 3. 

### 3.2. Synthesis

The data presented in the papers covered three broad research questions: (1) why do children think physical activity is important? (2) what are the factors that influence children’s physical activity? and (3) what do children like when it comes to physical activity? The themes and subthemes related to these questions are discussed below. Example quotations for all subthemes are provided in Table 4, Table 5 and Table 6. 

### 3.3. Why Do Children Think Physical Activity Is Important?

Across the studies, children highlighted several reasons for being physically active that centred around their own physical activity preferences and the benefits they associated with being active. Enjoyment was discussed in half of the studies (N = 13) [20,21,22,24,27,30,31,32,33,37,38,40,41] and was often attributed to specific activities (such as swimming or playing on the trampoline), places (such as the beach or skate park) or being active with specific people, particularly friends.

In five studies [21,24,25,26,36], children associated being active with the opportunity to be outside. This was seen to be particularly enjoyable in the school context as outdoor PE was highlighted as being a welcome contrast to other indoor lessons. As well as enjoying being active outdoors, the children felt that it also afforded educational opportunities for learning about nature. 

Across most of the studies, children discussed a variety of benefits they associated with being active [20,21,22,23,24,29,30,32,34,35,36,37,38], including improving and maintaining health and fitness. The children recognised both the immediate physical health and psychological well-being benefits of being active and demonstrated an awareness of the longer-term preventative association with certain diseases. Some children also felt being active was important for weight control. Children in four studies [23,24,30,38] recognised the benefits of being active for their mental well-being due to their feelings associated with being active. These positive feelings were attributed to increases in confidence from seeing their skills improve as well as the way being active helped them control their emotions and relieve stress. In one study [38], children also felt that the emotional release that being physically active gave them also resulted in improved behaviour. 

In many studies children highlighted that regular physical activity was important for their learning and development (N = 8) [22,23,24,27,33,34,37,40]. This was through having the opportunity to try to develop in different positions in team sports as well as practicing and building confidence in individual activities such as dance and gymnastics. Confidence, or a lack thereof, was also discussed across six studies [23,25,33,37,40,41] and was often linked to physical activity performance. When children perceived their own performance to be good, they felt their confidence increased. However, they also frequently compared their performance with other people which often led to feeling less confident in their own abilities. Children in four studies [21,24,38,40] felt that being physically active provided them with opportunities for social development. For many children, this was a key driver in determining their physical activity levels as they gained more enjoyment from being active with other people, particularly their peers, and engaging in different physical activities also provided opportunities for them to meet people and make new friends. 

### 3.4. What Are the Factors That Influence Children’s Physical Activity?

Across the included studies, children cited a range of different factors that they felt influenced their physical activity, including social connections with various social agents, messaging around physical activity in the media, and practical considerations including the cost of activities. 

### 3.5. Social Influences

Friends, family, teachers and coaches were all felt to have important positive and negative influences on physical activity engagement. Friend and peer relationships were discussed the most, with children in 15 studies [21,24,25,26,30,31,32,33,35,36,38,40,41,42,44] highlighting that they were more likely to be active when with their friends due to feelings of enjoyment.

This was the case across all physical activity contexts (see Table 5 for subsequent example quotations). Similarly, children in six studies [30,31,32,36,41,43] indicated that, due to living further away from their friends, they were able to be active with their friends more at school, and so they enjoyed school-based physical activity more than home-based activity. This extended to children feeling their physical activity levels outside of school were influenced by where their friends lived. Those who lived close to friends talked of being active with them in their free time, yet those who did not have friends living close by generally felt that they were less active outside of school. In five studies [24,32,34,37,41] children felt like friends had a positive influence on their physical activity due to the encouragement received from friends through both verbal communication that helped them to increase their confidence in their ability when it came to specific physical activities as well as through friends offering opportunities for them to try new physical activities.

Children identified that some interactions with peers could negatively impact their physical activity engagement. In seven studies [23,24,34,36,37,38,41,42], and particularly in the context of structured physical activity such as in PE or organised team sports, team and class dynamics were instrumental in determining children’s enjoyment of activity. Children felt that team dynamics in organised sports often involved a high level of peer pressure, and this meant their enjoyment of, and subsequent engagement in, the activity decreased. In the context of more informal physical activity, such as during school break and lunchtimes, other children’s bad behaviour sometimes made being active less fun [23,24,36,38,41,42] as bad behaviour meant certain activities were banned during recreational periods.

In two studies [32,36], children discussed popularity and friendship group differences in physical activity, emphasising the impact that friendship group norms could have on activity levels. Positively, children felt that their different friends enjoyed different activities, and this therefore encouraged them to be active in a variety of ways. However, the children also felt that differences between friendship groups in terms of what was perceived to be normal and ‘cool’ meant that there were distinct differences in activity levels between groups. These differences were particularly clear when looking at different gender groups. Whereas the children felt it was imperative to be physically active to be a popular boy, the popular girls were generally perceived to be less active and engaged in more sedentary social activities during school breaks. In one study [31], intimidation from older children was also felt to influence activity levels outside of school. Fears concerning older children were specifically related to being active in public parks, with a group of 10–11 year old children who were living in Scotland and from low–middle income families feeling intimidated by older children who also spent time in the park and made the younger children feel like they were not welcome there.

Whilst discussions around friend and peer influences on physical activity were present in most studies, the influence of family was also frequently discussed. Children in 11 studies [20,21,25,26,27,28,30,37,39,40,41] reported being active with family including parents and siblings. As with friends, most children felt that being active with their family was fun and central to their out-of-school activity. For some children, their family culture helped them to be more active due to the wider family behaviours and a self-labelled identity as an ‘active family’. For children in two studies [39,40] this translated as behavioural modelling where they engaged in the same activities as their parents and siblings. Children in four studies [28,34,37,40] felt they received a good level of support from parents in terms of maintaining their physical activity levels. This support manifested in several ways but generally involved some logistical support, such as being given lifts to and from training as well as parents attending their sports matches. Parent communication was also seen to influence physical activity engagement across four studies [28,33,34,40]. This was generally viewed positively, with parents expressing feelings of pride when the child performed well. However, some children expressed negative experiences, particularly when parents shouted at them when they did not perform as well as expected. Despite children in many studies reporting being active with members of their family, in five studies [27,28,30,37,40] children spoke of instances where parent behaviour was a barrier to activity. This was generally in the form of parental restrictions, and most of these children understood that the parental restrictions were due to perceptions of neighbourhood safety, but some children felt that their parents’ personal reservations about being active, such as body concerns related to swimming, meant they did not facilitate physical activity opportunities for their children.

Across several studies, and particularly in those exploring physical activity within the school or organised sport context, children also highlighted that the way they interacted with their PE teachers and sport coaches influenced their activity engagement. Coach behaviour and communication was discussed in five studies [23,28,33,34,45], with children feeling that specialist coaches both in and out of school generally encouraged them to be more physically active by pushing them to develop their skills in particular activities. However, in some cases, and particularly in the school context, communication with coaches was viewed as being less personal than with teachers, which deterred some children from engaging in the activities. Similarly, children in four studies [23,27,35,41] felt teacher behaviour and communication was instrumental in their activity levels, as PE teachers pushed them more in lessons, leading them to be more active during this time. However, being pushed to work harder meant some children did not enjoy PE and this led to them frequently giving excuses to miss lessons. Teachers encouraged activity by helping children to improve their skills through structured lessons (N = 3) [23,27,41] and children generally felt their teachers (both PE teachers and wider school staff) cared about them and their activity levels. Teachers’ own activity behaviours were also discussed in one study [35], with one child observing that teachers regularly drive to the local shop during school lunchtimes, thus having a negative impact on the child’s own motivation to travel actively.

In addition to their friends, family, teachers and coaches, children in two studies [27,35] also felt that the media influenced their activity levels. Children reported finding physical activity information from several print and digital media sources, including books, posters, television and websites. Some children discussed proactively seeking out this information to learn how to improve their technique in a particular activity or to learn how to work a particular part of their body. For others, information was viewed more passively, such as through noticing advertisements for schemes aiming to get children to be healthier. Some children also identified athletes that were prevalent within the media as role models.

### 3.6. Practical Influences

Several practical issues were felt to influence physical activity, including issues related to age-appropriate physical activity provision as well as potential barriers to activity such as cost and time. In five studies [27,35,37,39,40], children felt that the weather influenced their activity levels, although there was little consensus between children as to the pattern of this influence. For some children, rain and cold weather inhibited them from going outdoors or from travelling to school actively, either from their own choices or parental discretion, which led to a reduction in physical activity. Other children highlighted that they were still allowed outside in the rain and enjoyed running around when it was wet. The appropriateness and availability of physical activity provision and facilities was felt to influence physical activity levels by children in three studies [30,31,40]. Children discussed their preferences with regards to physical facilities, such as enjoying being active in parks with multiple facilities, as well as the availability of provision, such as attending multiple physically active clubs after school. However, some children also reported that they stopped enjoying clubs they used to attend as they felt they were no longer appropriate for their age. Children in three studies [28,37,40] spoke of having to stop attending activities they enjoy, such as swimming and horse riding, due to the associated cost of these (both the cost of attendance and of equipment). Children highlighted time as having an influence on their activity levels in four studies [27,37,40,45]. Some children identified that they were more physically active at the weekend as they had more free time and fewer of the competing demands that are present during the week. In one study that focused on ethnic differences in physical activity, religious practices were highlighted as a key barrier to being active. This was due to the necessary and scheduled time that was required to dedicate to prayer and going to the place of worship, in this case the mosque, which meant that these children were often unable to participate in physical activity after school, and particularly to participate in organised clubs. 

### 3.7. What Do Children Like When It Comes to Physical Activity?

Throughout many of the studies, children discussed what they did and did not like with regard to specific physical activities as well as the features of activities that they felt encouraged them to be more active.

### 3.8. Activity-Related Preferences

Across the studies, children expressed a variety of preferences regarding physical activities they enjoyed. In 10 studies [19,23,24,25,30,37,40,41,43,45], children preferred specific structured activities such as football, gymnastics and boxing.

Activities were generally preferred when carried out with friends or because of the child’s perceived ability in the activity. As well as structured activities, children in 10 studies [20,21,24,25,30,30,32,36,37,39] also enjoyed more informal outdoor physical activities. These were largely activities participated in during leisure time, such as playing with friends on their bikes. Some participants also mentioned that owning a dog helped them to engage regularly in outdoor physical activity due to walking them regularly. Active travel was highlighted by children in two studies [35,40] as something they enjoyed that contributed to their physical activity levels. These children reported enjoying walking or cycling to school as it gave them additional time with their friends.

Regardless of their preferences for either structured or informal physical activity, most children enjoyed participating in a range of physical activities and the variety that this gave them (N = 10) [20,21,22,23,27,32,36,37,41,42].Children were likely to become bored when an activity was repeated too frequently. In six studies [20,22,23,24,29,30,31], children communicated a preference for being active over being sedentary. This was particularly the case at school, where PE lessons were compared favourably to classroom-based lessons. However, in seven studies [21,27,28,37,39,40,41], children expressed a preference for indoor activities that were primarily sedentary during their leisure time. For some children, their natural preferences inhibited their physical activity levels, whereas others felt they understood the value of going outdoors and being active despite naturally wanting to engage in more screen-based activities.

### 3.9. What Helps Children Be More Active?

Regardless of individual preferences for activities, across the included studies there were several features of activities that children reported enjoying and that made certain activities more appealing. The most often discussed of these was the inclusion of creative physical play, defined as children taking the lead in developing and initiating activities (N = 7) [20,21,23,34,36,37,43]. Such child-led activities were felt to encourage children in general to be more active, particularly when they were provided with equipment to use to create their own games. Relatedly, an element of choice with regards to activities was felt to encourage more children to be active by children in four studies [23,28,34,41]. Whilst children in many studies wanted more freedom and choice, in some studies (N = 2) [22,34] the children felt that some rules made being active more fun by providing structure. This was particularly the case when children were not behaving and is related to the social influences that children discussed in terms of team and class dynamics.

Children in six studies [27,33,34,35,41,42] enjoyed activities that included an element of competition, especially between close friends, and thought it encouraged them to be more active. In one study [35], children suggested that adding a competitive element to activities, such as a whole-class competition, might initiate engagement in new activities (e.g., active travel) which they might potentially continue with long term. Across six studies [23,24,33,34,35,42], activities with a challenge aspect were felt to be key to helping children to be more active. Seeing themselves improving and the sense of achievement that was associated with challenging activity was viewed as positive by the children. However, this challenge needed to be balanced with the children’s own perceptions of competence, as these were felt to be central to physical activity engagement in three studies [40,41,44]. Children reported being more likely to engage in activities they felt that they were good at, and that not having the necessary skills (e.g., riding a bike) was a key barrier to behavioural initiation. In many cases, perceptions of competence originated from acceptance and recognition from others, particularly their peers.

## 4. Discussion

Across the studies included in this review, children cited a range of reasons for being physically active, including enjoying being physically active and being outdoors. Children in over half of the studies also highlighted benefits to both their physical and psychological health, in line with the extensive evidence of improved health associated with physical activity [1,2]. Additionally, many children also discussed the developmental opportunities afforded to them through physical activity in terms of physical and social skills and behavioural management. Historically, it has been debated whether children have a good understanding of the benefits of physical activity, with many suggesting that children’s understanding differs somewhat from that of adults [47]. It has also been suggested that physical activity messaging for children should not focus on physical health improvement or prevention, as these outcomes are too abstract and distal for this population [48]. However, the qualitative data from the studies included in this review highlight that, in general, children are aware of the multitude of short- and long-term benefits from physical activity. These differences could reflect an increased societal focus on health in which children are more frequently exposed to physical activity and health-related messages through mass media campaigns [49].

There has been extensive qualitative work exploring the roles of significant others (friends, family and teachers) in promoting or hindering children’s physical activity engagement. Previous qualitative reviews in this context focussed on identifying the barriers and facilitators to children’s physical activity, and these identified parental support as a key facilitator of activity [15,17]. However, in the more recent qualitative studies included in this review, whilst parental behaviours were discussed, relationships with friends and peers appeared to play an equally important role in determining children’s activity. This is likely due to the frequency with which children are active with friends and the social dynamics within friendship groups. The impact of peer relationships may have become more pertinent in recent years due to the increasing use of smartphones and social media in children [50]. There have been several recent interventions targeting physical activity in adolescents through peer relationships [51,52], and initial findings indicate that these strategies are effective [53,54]. The findings of this review suggest that peers may be key to increasing younger children’s physical activity, with the children highlighting that they enjoy being active with friends and expressing that peer behaviours and communication can both facilitate and hinder their activity engagement. Analysis of quantitative data from UK primary-school children shows that physical activity levels are similar within friendship groups, thus indicating that friendships influence physical activity [55,56]. Subsequently, peer-based interventions for younger children may hold promise for increasing primary-school children’s physical activity.

The more recent qualitative evidence has also raised questions regarding the role of teachers and coaches in children’s activity, and across the studies included in this review there were several teacher and coach behaviours that may be facilitative or detrimental to children’s physical activity engagement [23,27,33,34,35,37,38,41,42]. Perceptions of teacher and coach interactions were mixed, with many children identifying behaviours that had a negative impact on their desire to engage in physical activity. Quantitative evidence has found that interventions aiming to increase children’s physical activity are more effective when they target multiple contexts (e.g., school and family) and therefore multiple social agents [57]. The children’s perspectives suggest there may be value in involving teachers and coaches in interventions aiming to increase children’s physical activity, with a particular focus on encouraging more supportive behaviours and communication.

Over the last two decades, primary-school children have had increasing access to media devices, with up to 75% owning a smartphone or tablet at age 8–11 years [50]. This review suggests that media, including television and internet use, can have influences on physical activity levels through viewing advertisements and presenting role models. Whilst much of the literature has focused on the negative impact that media use and screen-viewing can have on activity levels [58], a recent exploration of how young people in the UK use digital technologies indicates that there is also the potential for such activities to be beneficial, with most young people finding health information online [50]. There is also some initial evidence indicating that interactive media (e.g., social media) can promote healthier lifestyle behaviours in children and adolescents [59] through the use of active video games to engage children in physical activity [60,61,62].

In line with the findings of previous reviews [15,17], the enjoyment of activities, safety perceptions, competition and the structure of physical activity were key themes. Whilst parents’ perceived environmental safety has long been found to be associated with children’s physical activity [63,64], the focus on data from children themselves highlights that children’s own perceptions of safety may also be pertinent. Specifically, in one study, children aged 10–11 years spoke of feeling intimidated at their local park due to the presence of older children [31]. Previous studies have emphasised the importance of the age-appropriateness of activity provision, particularly in relation to perceptions of activities as being for younger children [65]. However, this review indicates that it may also be important to consider the appropriateness of facilities and activity locations for primary-aged children. Further qualitative work with children is needed to identify the places and activities which they deem to be acceptable and appealing.

The current findings corroborate other studies that have highlighted practical considerations for intervention design and development, including weather, cost and time. Most studies have involved children of white ethnicity, with only five papers included in this review involving children from Black Caribbean, Black African, Indian, Pakistani and Bangladeshi communities [25,27,33,39,44]. The findings of these papers highlight important considerations for children in Black and Asian minority ethnic groups, such as the timings of physical activity provision, which are often not accessible for children from some cultures due to religious-practice commitments. This may be particularly pertinent to changing physical activity levels at the population level, as some evidence suggests children of ethnic minorities (including Bangladeshi and Pakistani communities) engage in less activity than similar white European children [66,67]. However, current national physical activity monitoring in the UK, such as the Health Survey for England [68], are unable to show population-level differences in physical activity by ethnic group due to very small samples of participating children from Black and Asian minority ethnic groups. There is a need to engage children from these communities in physical activity research to understand their needs in terms of strategies to increase physical activity.

This review provides some direction in terms of key aspects to include in physical activity interventions through exploring what primary-school children like in terms of physical activity (see Table 7). Children across several studies highlighted enjoying a variety of activities, a preference for child-led activities and creative physical play that also offers structure that enables them to see themselves developing and improving, indicating that these should be integral features of future interventions.

This review brings together findings from a diverse body of qualitative papers which presented two key methodological challenges when answering the research questions. First, the papers reviewed adopted a range of different methodologies, including interviews, focus groups, photovoice, and write and draw. Whilst all methods elicited insightful qualitative data from the children, papers that used multiple methods appeared to cover a broader range of issues than those using single methods, particularly those adopting more creative, multisensory methods (e.g., photography, write and draw) alongside traditional qualitative interviews or focus groups. Secondly, the central aim of this review was to synthesise what is already known about primary school children’s perspectives on physical activity. Primary school typically encompasses ages 5–11 years, yet the majority of studies included in this review focussed on children in the final years of primary education (9–11 years), with only one paper involving six year olds and very few quotes from these younger children presented in the original papers. Given that physical activity begins to decline from the start of primary school (5), there is a need for further research with younger children to inform intervention development as earlier intervention may have the greatest potential for maximising physical activity engagement (6). One of the key challenges in including younger children in research is encouraging active participation whilst ensuring that the expression of their thoughts and ideas is not inhibited [69]. Previous research has highlighted the benefits of adopting multiple methods in physical activity research with children, emphasising the use of combined methods for gaining insights and understanding beyond those that can be accessed via a single method [70]. The use of more creative methods, such as photography, alongside interviews and focus groups might enable researchers to explore child perspectives on physical activity more thoroughly and to encourage active participation from younger children without inhibiting the expression of their ideas.

There are, however, limitations of this review. First, as previously mentioned, the papers were diverse in terms of both the methodology used and the context explored, which presents challenges in synthesising literature that is constructed within fundamentally different philosophical perspectives. Whilst the thematic-synthesis method used to review these papers has somewhat accounted for this by focussing only on the quotes from children rather than researcher interpretations, the data presented in the papers may differ accordingly. The choice to focus only on quotes from children means that our analysis is restricted by the choices made by the authors of the original papers in terms of what quotes to present. We therefore want to emphasise that the aim of this review is concerned with ascertaining what is already known in the literature rather than aiming to provide a comprehensive overview of children’s views. However, we acknowledge that the themes here may only be the tip of the iceberg and emphasise a need for more open data reporting within qualitative research. Finally, whilst the majority of studies included in this review were deemed to be of a moderate-to-high quality using the CASP tool, the quality appraisal highlights the need for better consideration of the role of the researcher in qualitative research with respect to their position and skill in research with children. This is particularly important in research with children, where the potential power dynamics between children and researcher could have an impact on how easily the children feel able to express their opinions.

## 5. Conclusions

Children have a broad understanding of the benefits of physical activity, including for enjoyment, health and fitness and social development. Family, friends and teachers are key social agents that influence physical activity, and influences from the media and more practical issues such as cost and time are also important. The studies highlight the diversity in children’s preferences for activity, but most children like opportunities for creative play as well as some structure. By focussing on the perspectives of the children themselves, this review emphasises important issues that future strategies to increase physical activity should consider to ensure that programmes are appropriate and acceptable to the target population.

## Figures and Tables

**Figure 1 ijerph-19-03993-f001:**
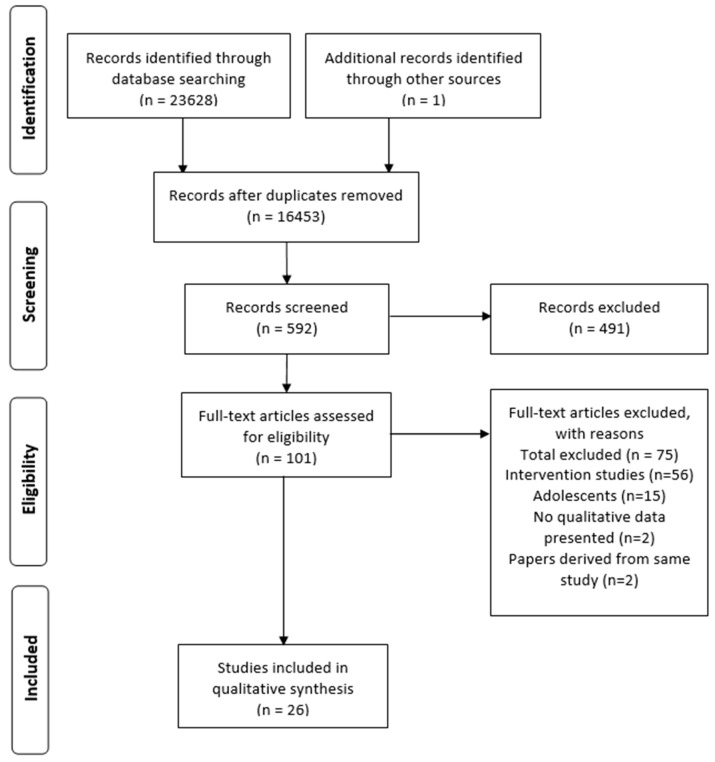
A PRISMA flow diagram outlining the search process.

**Table 1 ijerph-19-03993-t001:** Search strategy used to identify qualitative studies on children’s physical activity since 2004.

Search terms (Terms and search operators varied slightly according to database guidelines).	
**Concept 1** (population)	child* OR “primary$school” OR “elementary$school”
**Concept 2** (focus)	physical* activ*” OR sport* OR run OR running OR swim OR swimming OR walk OR walking OR cycle OR cycling OR dance OR dancing OR play OR playing OR “physical education”
**Concept 3** (design)	qualitative OR photovoice OR participatory OR interview OR “focus group” OR ethnograph *
**Search limits**	2004–2019
**Databases**	CINAHL, EThOS, IBSS, Medline, PsycInfo, Scopus and Web of Science
**Date of final database search** (conducted by LEC)	27 January 2022
**Supplementary search strategies**	Searching reference lists of relevant studies, searching research citing relevant studies, searching other work by authors of relevant studies

* is a common boolean operator within systematic review searching. In this case search terms starting with ethnograph would be found (e.g. ethnograph, ethnography, ethnographic, ethnographical).

**Table 2 ijerph-19-03993-t002:** Quality appraisal using the CASP tool.

Source	Section A						Section B			Section C
	Aims	Sound Method	Appropriate Design	Recruitment	Data Collected Appropriately	Role of the Researcher	Ethics	Data Analysis	Findings	Value
Ashbullby et al., 2013 [20]	✓	✓	✓	✓	✓	✕	-	✓	✓	✓
Brockman et al., 2011 [21]	✓	✓	✓	✓	✓	✕	✓	✓	✓	✓
Dismore et al., 2011 [22]	✕	✓	-	✓	-	✓	✕	✓	✓	✓
Domville et al., 2019 [23]	✓	✓	✓	✓	✓	-	✓	✓	✓	✓
Everley et al., 2017 [24]	✓	✓	✓	-	✓	✓	✓	✓	✓	✓
Everley et al., 2019 [25]	✓	✓	✓	-	-	✕	✕	✓	✓	✓
Everley, 2020 [26]	✓	✓	-	✓	✓	✓	✓	✓	✓	✓
Eyre et al., 2015 [27]	✓	✓	✓	✓	-	✕	✓	-	✓	✓
Furusa et al., 2021 [28]	✓	✓	✓	✓	✓	-	✓	✓	✓	✓
Gosling et al., 2008 [29]	✓	✓	✓	-	✓	✓	✓	✓	✓	✓
Hayball et al., 2016 [30]	-	✓	-	✓	✓	✓	-	-	✓	✓
Hayball et al., 2018 [31]	✓	✓	-	-	✓	✕	-	✓	✓	✓
Jago et al., 2009 [32]	✓	-	-	✓	✓	✓	✓	✓	✓	✓
Keegan et al., 2010 [33]	-	✓	✓	✓	✓	-	✓	✓	-	✓
Keegan et al., 2009 [34]	✓	✓	✓	✓	✓	✕	-	✓	✓	✓
Kirby et al., 2009 [35]	✓	✓	✓	-	✓	✕	✓	✓	✓	✓
Knowles et al., 2013 [36]	✓	✓	✕	✓	-	✕	-	✓	✓	✓
Mackintosh et al., 2011 [37]	✓	✓	✓	✓	✓	-	✓	✓	-	✓
Medcalf et al., 2011 [38]	✕	✓	-	-	✓	✕	✕	✓	✓	✓
Murphy et al., 2021 [39]	✓	✓	✓	✓	✓	✓	✓	✓	✓	✓
Noonan et al., 2016 [40]	✓	✓	✓	✓	✓	-	✓	✓	-	✓
Parker et al., 2018 [41]	✓	-	✓	-	✓	-	-	✓	✓	✓
Powell et al., 2016 [42]	✓	✓	✓	✓	✓	✓	✓	✓	✓	✓
Powell et al., 2019 [43]	✓	✓	✓	-	✓	✓	✓	✓	✓	✓
Rawlins et al., 2013 [44]	✓	✓	✓	✓	-	✓	✓	✓	✓	✓
Spotswood et al., 2021 [45]	✓	-	✓	-	✓	✕	✓	✓	-	✓

Note: ✓ denotes yes, ✕ denotes no and - denotes cannot tell.

**Table 3 ijerph-19-03993-t003:** Study and participant characteristics for the studies included in this review.

Authors	Publication Type	Context	Year Data Collected	Location	Qualitative Method	Number of Focus Groups/Interviews (Duration)	Number of Children	Age Range	Gender	Data Analysis Framework
Ashbullby et al., 2013 [20]	Journal article	Physical activity at the beach	2011	Devon and Cornwall	Interviews	20 (15–30 min)	20	8–11 years	10 boys, 10 girls	Thematic analysis
Brockman et al., 2011 [21]	Journal article	Active play	2009	Bristol	Focus groups	11 (30–40 min)	77	10–11 years	22 boys, 50 girls	Thematic analysis
Dismore et al., 2011 [22]	Journal article	Physical education	-	-	Focus groups and open-ended written responses	19 focus groups (Not reported)	790 in questionnaires, 86 in focus groups	7–14 years	-	Thematic analysis
Domville et al., 2019 [23]	Journal article	Physical education	-	North West England	Focus groups	8 (30–5 min)	47	7–11 years	23 boys, 24 girls	Thematic analysis
Everley et al., 2017 [24]	Journal article	Physical activity	-	South England	Drawing and interviews	83 (15 min)	83	6–10 years	33 boys, 60 girls	Drawing and thematic analysis
Everley et al., 2019 [25]	Journal article	Physical activity	-	South England	Imagination, drawing and interviews	29 (not reported)	29	5–6 years	14 boys, 15 girls	Content analysis
Everley et al., 2020 [26]	Journal article	Physical activity	-	South England	Drawing and interviews	113 (not reported)	113	5–10 years	-	Critical visual methodology framework and thematic analysis
Eyre et al., 2015 [27]	Journal article	Physical activity	-	Coventry	Focus groups	5 (40–50 min)	33	7–9 years	16 boys, 17 girls	Thematic analysis
Furusa et al., 2021 [28]	Journal Article	Sport	-	-	Focus groups	9 (53–66 min)	32	8–11 years	9 boys, 23 girls	Thematic analysis
Gosling et al., 2008 [29]	Journal article	Physical activity	-	North West of England	Focus groups	4 (60 min)	32	9–10 years	16 boys, 16 girls	Thematic analysis
Hayball et al., 2016 [30]	Doctoral thesis	Outdoor physical activity	2015	Scotland	Visual images (photography and drawing), interviews and focus groups	3 focus groups (57–98 min) and 11 interviews (22–53 min)	25 in image generation, 20 in focus groups and interviews	10–11 years	12 boys, 13 girls	Grid categorisation for images (by child) and thematic analysis
Hayball et al., 2018 [31]	Journal article	Outdoor physical activity	2014	Glasgow	Visual images (photography and drawing) and focus groups	3 (45–120 min)	15	10–12 years	5 boys, 10 girls	Grid categorisation for images (by child) and Thematic analysis
Jago et al., 2009 [32]	Journal article	Physical activity	2007	Bristol	Focus groups	17 (30–45 min)	113	10–11 years	54 boys, 59 girls	Content analysis
Keegan et al., 2010 [33]	Journal article	Sport	-	-	Focus groups	8 (45–65 min)	40	7–11 years	21 boys, 19 girls	Content analysis
Keegan et al., 2009 [34]	Journal article	Sport	-	-	Focus groups	12 (45–65 min)	79	9–18 years	59 boys, 36 girls	Content analysis
Kirby et al., 2009 [35]	Journal article	Active travel	2006–2007	Scotland	Focus groups	13 (15–20 min)	66 (25 primary school children)	10–13 years	29 boys, 37 girls (10 boys, 15 girls at primary school)	Content analysis
Knowles et al., 2013 [36]	Journal article	Physical activity during recess	2003–2004	Northwest England	Write and draw	(30–45 min)	299	7–11 years	-	Content analysis
Mackintosh et al., 2011 [37]	Journal article	Physical activity	-	North West England	Focus groups and interviews	13 (30–45 min)	60	9–10 years	24 boys, 36 girls	Pen profiles (YPAPM)
Medcalf et al., 2011 [38]	Journal article	Physical education	-	-	Interviews	31 (not reported)	6	-	All boys	Inductive reasoning
Murphy et al., 2021 [39]	Journal article	Physical activity	2015–16	Coventry	Draw, write, tell and interviews	26 (14–59 min)	26	9–10 years	11 boys, 15 girls	Framework analysis
Noonan et al., 2016 [40]	Journal article	Out-of-school physical activity	-	North west England	Write, draw and show-and-tell groups	7 (40–55 min)	35	10–11 years	16 boys, 19 girls	YPAPM and thematic framework analysis
Parker et al., 2018 [41]	Journal article	Physical education and out-of-school physical activity	-	Ireland	Write, draw and focus groups	11 (not reported)	135 (write and draw) 34 (focus groups)	8–11 years	22 boys, 12 girls in focus groups, 86 boys, 49 girls in write and draw	General inductive approach
Powell et al., 2016 [42]	Journal article	Physical activity during recess	2013–2014	West Midlands	Focus groups	10 (30 min)	80	7–10 years	47 boys, 33 girls	Interpretive phenomenological analysis
Powell et al., 2019 [43]	Journal article	Physical education	2014–2015	West Midlands	Focus groups	10 (not reported)	80	7–9 years	42 boys, 38 girls	Interpretive phenomenological analysis
Rawlins et al., 2013 [44]	Journal article	Physical activity	2008–2009	London	Focus groups	13 (45 min)	70	8–13 years	31 boys, 39 girls	Thematic analysis
Spotswood et al., 2021 [45]	Journal article	School-based physical activity	2017	England	Focus groups and paired interviews	Unknown number of focus groups, 6 paired interviews (15–30 min)	25	Not reported, primary school age	-	Framework analysis

**Table 4 ijerph-19-03993-t004:** Illustrative quotations and references reporting each of the subthemes for ‘Why do children think physical activity is important?’.

Themes and Subthemes	Participant Quotations from Primary Studies (Child Gender where Known; Reference)	N Studies (%)
Enjoyment of activity	“Well this is a park, it’s like a park but then it has like a really like good size wood next to it and I like to go there and play with my friends because it’s fun and sometimes you don’t know where you’re going, which I also think is really fun. And you sort of just have to work your way around it like a maze, it’s really fun to play with, with your friends, and I enjoy going there” Girl [31].	13 (50%)
Health and fitness	“It stops you getting cancer. Diseases and diabetes”. [27] “If you didn’t do exercise then you would end up being fat”. [27] “Well I like it, like if you’ve just been in a rubbish lesson it gives you chance to run it off, you think of something else and just run it off … just get to run around, burn some energy off” Boy [38].	12 (46%)
Getting outside	“It doesn’t get boring because we have to do different subjects. Coz we’ve got PE indoors which is just like one class, just use the equipment and then we’ve got PE outdoors where we can like run free and like, be with friends in other classes” Boy [22]. “It gets you sort of aware of your surroundings and […] it’s a great way to look at nature and stuff as well and just realise how cool the world is basically” Boy [20].	5 (19%)
Feelings associated with being active	“It just feels nice when you hit the ball … like, because there are loads of them strings, when you hit it, it just feels nice” Boy [38]. “This is me on my trampoline having lots of fun. I like playing on my trampoline; it makes me feel alive” Boy [24].	4 (15%)
Learning and development	“They say, may be [playing well] that’ll put you in this place and you’ve never been there before’’ Boy [34]. “Well, it makes you feel comfortable because you know that if you get something wrong, they’re [instructor] just going to help you and try again, and they’ll tell you to try again, and then eventually when you get it, they’ll say that we’ve made progress and still help us build up the confidence” Girl [23].	8 (30%)
Confidence	“I don’t play with the boys in my class—I don’t do football at home cos I don’t know what to do really—I can’t get the ball and it gets all stressy ‘cos there’s just no time to decide what to do with it so I don’t play—I go in the wild garden” Boy [25]. “There’s loads and loads of black-belts in the room. All staring at you, doing your thing. So you’re practicing and you don’t know, you don’t know whether you’ve passed or not and you’re not sure of one move, and I just feel a bit weird if I don’t know that set move, and if I’m gonna do it right or wrong” Boy [34]. “I like swimming because I’m a fast swimming” Boy [40].	7 (27%)
Social development	“Like when I’ve playing football in my back garden just on my own, it’s not as fun, because you can’t pass to anyone except for the wall” Girl [40]. “Children can meet other people [at the beach]” Girl [20].	3 (12%)
Improves behaviour	“Yeh it makes me not as like, not as like naughty or stuff … so I’m just like chilled out if you know what I mean” Boy [38].	1 (4%)

Note. Quotes are presented with gender of child, where available, and paper reference.

**Table 5 ijerph-19-03993-t005:** Illustrative quotations and references reporting each of the subthemes for ‘What are the factors that influence children’s physical activity?’

Themes and Subthemes	Participant Quotations from Primary Studies (Child Gender, Where Known; Reference)	N Studies (%)
**Social influences**		
**Friends and peers**		
Being active with friends	“In school you’re with your friends and when you’re at home you’re, like, your friends aren’t with you. When your friends are there with you playing it, it makes it more fun” Boy [41].	15 (58%)
Children’s bad behaviour	“Sometimes people are listening, and then you get other people that just think about themselves and they don’t think about the team, and they never listen, so then like say someone who’s like talking to someone, [the instructor] would go, “Oh, everyone, I’ll tell you again, and I’ll tell you again”, and it just gets really boring, because we’ve listened, but they haven’t” Girl [23]. “We used to play football every single day but then it got banned because people kept kicking each other” Boy [37].	8 (31%)
Team and class dynamics	“People always shout at you, like not for doing it right, and then people on my team, [they say] “Oh, come on. Why are you out?” and things like that…it’s like they always hit it [the ball], and you never do, so like you feel a bit, you feel as if you’ve let your team down…But like when you’ve got a positive team, and like they’re really nice, they’ll keep cheering you on, and you’ll keep making you do more, like to believe in yourself” Boy [23]. ‘‘Yeah because like in swimming [relay] like if you’re the last one to go and like all your team-mates have made you be in front then they’re like depending on you and that makes you feel like. pressure’’ Girl [34].	7 (27%)
Proximity of friends	“Well my best friend lives opposite me … and my other two friends don’t live far so I just play with them” Boy [21]. “Me and Georgina live on the same like street and it’s like a cul de sac so not many cars go so we play out quite a lot” Girl [37].	6 (23%)
Friend encouragement	“I would tell my friends and then some of them will be really supportive and like try to help me to reach the goal” Girl [37]. “my friend [child’s name] encouraged me to go to cubs because she said she was going and it was really fun” Girl [32].	5 (19%)
Popularity and friendship group differences	“I would say I think it helps your popularity in the boys’ group to be physically active … ” Girl [32]. “Well, all the different groups of friends that I have they are all different so it’s kind of you get a different variety of friends and different kinds of people you get to know and so it’s kind of like sharing all the different sides of you that you have … ” Girl [32].	2 (8%)
Older children and intimidation	Boy 1: “You get the ones at the Peel who are in, colourful like football strips and they’re just running around happy”. Boy 2: “And you get the ones like what I was talking about at the parks, just in grey hoodies, and you can’t see their face and then they’ve got jogging bottoms on and you just, it’s like they just don’t want you to see that they’re hanging around there, but they kind of do because they want people to see that they’re cool”. [31]	1 (4%)
**Family**		
Being active with family	“Sometimes I take my sister, my dad takes me and my sister over there to play on the activity trail…Yeah, it’s fun. More adventurous”. Boy [30]. “This is me playing outside with my Daddy (stepfather)– I also play in the park with my Daddy Daddy (biological father)—I run outside a lot ‘cos I see both my Daddies outside” Girl [25].	11 (42%)
Support from parents	“Because my mum and my dad try to get to every race or football match that I’ve done, and they always come and support me wherever I am”. Boy [40]. “If you’ve had like a really tough day at work and you came back and your child wanted to go to swimming practice or anything and you couldn’t be bothered to go, you’ve still got to take the child because they might actually turn out to be an Olympic swimmer … your parents have got to believe in you”. Boy [34]. “‘Well they pay for you to play football and they always help you and support you in everything you are doing”. Boy [28].	4 (15%)
Parents as barriers	“Sometimes my mum doesn’t let me out, we use to play with these children but their house got robbed so we couldn’t play out anymore’’ Girl [27]. “I want to go swimming with my Mum won’t go because she thinks she’s overweight so she won’t go in the pool with us’’. Boy [31].	5 (19%)
Parent communication	“Even when it’s obvious that you’re not gonna win they say ‘Do your best, carry on. Don’t give up!” and then afterwards they’re like ‘Well done! You played really well’, so you feel like you haven’t done so bad” Girl [33]. “they [parents] are very passionate and they just want you to improve, but […] it’s not very nice to hear them when they are shouting” Boy [28]. “I did it myself…I just do it by myself then I don’t have to get moaned at by my mum when I do it wrong” Girl [27].	4 (15%)
Family behaviours	‘‘If it’s quite a big reward, like a new Playstation game, and you like miss, you’re like really upset with yourself. it might have been your only chance to get it. And you’ve missed it’’. Boy [34]. “Because they’re all really involved in sport and going out everywhere and stuff” Girl [40].	2 (8%)
**Teachers and coaches**		
Coach behaviour and communication	“He laughs with you and makes you motivated and it’s like he’s a nice person it’s just that he wants us to win he wants us to do better’’. Boy [34]. “Because they [specialist coaches] just go like, “You”, or “You in the red bib” or like, “Number Seven”. Like learn my name. I don’t like getting called number seven or you in the red bib” Girl [32].	5 (19%)
Teacher behaviour and communication	“Sometimes she [teacher] makes us do more like a bit harder physical stuff, and sometimes not everyone likes to do it, so a lot of people get grumpy and things. And they start like not wanting to join in, and they start saying like they feel ill, just so they can get out of it” Girl [32]. “Basically all the teachers use their cars … our teacher when she goes to [local shop], that’s right over there, during school time she always takes the car”. Boy [35].	4 (15%)
Teachers encourage activity	“[PE teachers name] … make sure that you are good … make sure that we enjoy them … at lunch we play a skipping game and [teachers name] helps me to skip, he holds the rope” Girl [27].	3 (12%)
**Media**		
Media influences activity	“Things like today [active travel transition project], and there is posters everywhere, and there is always something going on to show you how to be more healthy … eating or being more active”. Boy [34]. “ … I saw this exercise programme it was called keeping your arms, your bum, your hips healthy … it showed you how you keep your bum and your legs fit. It was to keep the muscles on your bum” Girl [27].	2 (8%)
**Practical influences**		
Weather	“Sometimes we’re just stuck inside when it’s raining and I’d like to go to places” Girl [37]. “I get a taxi when it is really cold or I will walk in sometimes. My coat does have a hood but it doesn’t stay up so that is why I get a taxi”. [27] “When it’s raining, I will just sit down and watch TV. When it’s shiny [sunny] I will go outside, ride my bike, calling friends outside to play.” Boy [39].	5 (19%)
Provision and facilities	“When I go to my grandma’s, I’ve not really got much space to play because my grandma isn’t fi t enough to do the garden, [so] we can’t play in there and the road—it gets used a lot. There’s quite a few people that live on the way so I don’t really like to play out much.” Boy [29]. “I used to go to dance, street dance, jazz, ballet, gymnastics, but it stopped, so I went to a different one for a couple of week, and I didn’t like it, so I stopped” Girl [40].	3 (12%)
Time	“When I want to play out there’s never time cause we have to go shopping” Girl [37]. “I think maybe in the weekend when I have more time…so I tend to do like do more, like run around or go cycling maybe, in the weekend” [Girl; [40] “I have to go mosque and I will get told off so I have to mosque, which stops me from doing my swimming” Girl [27].	4 (15%)
Cost	“Like for horse riding you need to jodhpurs, the boots and the whip and the hat, because if you don’t have a hat you just can’t go, but then it’s for like horse riding, it’s like fifty pounds a week, so it’s really expensive. And for swimming, because I go for two hours it’s like sixty pounds, because I go four times a week” Girl [40].	3 (12%)

Note. Quotes are presented with gender of child, where available, and paper reference.

**Table 6 ijerph-19-03993-t006:** Illustrative quotations and references reporting each of the subthemes for ‘What do children like when it comes to physical activity?’.

Themes and Subthemes	Participant Quotations from Primary Studies (Child Gender, Where Known; Reference)	N Studies (%)
**Activity-related preferences**		
Outdoor physical activities	“Um like going around the streets on our bikes and stuff”. Boy [21]. “Cause we have a dog it’s quite easy cause he needs to go out it kinds of makes us go out and take him for a walk and get some exercise”. Girl [37].	14 (54%)
Structured activities	“This is me doing gym … ‘cos it’s my favourite thing and my friends seem to think I’m good at it”. Girl [24]. “What I like doing at playtime is playing football with all my friends”. Boy [42].	10 (38%)
Range of physical activities	“I like the trim trail because it’s different stuff, at first you’ve got to hang on and then you give your arms a break and you’re balancing and it’s all sorts of different things”. Boy [42]. “Like, if you’re starting a new one [physical activity] and like, if you just do it for a week it’s good but then when [the teacher] goes and starts doing it for ages that’s when it gets really boring”. Girl [41].	11 (42%)
Indoor activities	“ … I mean, if you stay at the house just doing nothing all the time and just going on like the computer … you’ll … just come home and just do it again but you need to actually get out and do something with your friends… it keeps you active and you can have fun”. Boy [32]. “[if] someone says there’s a programme on tonight and I was going to go out on my bike or go for a walk when I get home I’d rather watch the TV”. Girl [37].	7 (27%)
Active travel	“You get to spend more time with your friends if you are walking or cycling with them’’. Boy [35].	2 (8%)
Preference for being active	Girl 1: “Because you’re not sitting in lessons staring at the board all the time”. Boy 2: “Yeah and writing and that”. Boy 1: “You’re up and you’re doing something. You’re not just sitting there writing. You’re doing more things”. [22]	6 (23%)
**Activity features that help children to be active**		
Creative physical play	“We could go on the field and do whatever we want, and get skipping ropes on it, and have hula hoops, and bats and balls, and all them. That would just be really, really fun if we could have a freestyle week”. Girl [23].	7 (27%)
Competition	“Sometimes like if you fall out with them a bit and they say like ‘I bet you can’t do it’ then that can make you want to try hard and go and do it more, to prove them wrong. Even if you’re like best of friends it can turn to rivalry”. Boy [34]. “You could have a competition where you have a prize, and if they [students] did it [active travel] and found out they liked it, they might do it more often”. Girl [35].	7 (27%)
Challenge	“I don’t know why, I just like running…fresh air…it’s more challenging when it’s raining and it’s more natural”. Girl [24].	6 (23%)
Choice	Girl 1: “I reckon we could get to vote for which sport we do, and the [instructors] still get to choose every sport, and then we vote, and which one has the most votes we get to do”. Boy 1: “Yes, and you don’t get to choose what you’re doing”. Boy 2: “Like [name] said, we would vote on what sport we should do, because I think it’s like everyone’s opinion counts, saying what they want to do, not just the [instructors’]”. [23]	4 (15%)
Competence	“only the best get chosen and then if you’re not very good it’s, like, you’re not wanted” Girl [44].	3 (12%)
Rules make it fun	“I don’t think he’s strict enough. The other people are trying to concentrate, but then you get these like really naughty people who are trying to like mess up the lesson. and he doesn’t do anything!’’ Girl [34].	2 (8%)

Note. Quotes are presented with gender of child, where available, and paper references.

**Table 7 ijerph-19-03993-t007:** Key findings and recommendations for future strategies to increase primary school children’s physical activity engagement.

Finding	Recommendation
Children view friend and peer relationships as fundamental to their physical activity engagement.	Strategies should harness the power of these influential peer relationships and incorporate peer-focussed elements alongside other intervention components.
Regardless of individual activity preferences, children like opportunities to try a range of physical activities and enjoy the variety this brings.	Across all contexts, strategy leaders should ensure that there is a variety of activities on offer to cater to different physical activity preferences.
It was felt that opportunities for child led activities and creative physical play would encourage more children to engage in physical activity.	Sessions that encourage children to design their own physical games may help more children to be active. It is important that these are child-led but supervised by an adult to ensure that team and class dynamics do not undermine engagement in the games.
Children value structure in their activities as this helps them to monitor their development.	Sessions should balance child-led activities with some structured elements to help children improve and develop their physical skills.

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
