# Peer review of "Children’s Voices in Physical Activity Research: A Qualitative Review and Synthesis of UK Children’s Perspectives"

_ijerph, 2022, doi:10.3390/ijerph19073993_

Round 1

Reviewer 1 Report

This review is likely to be of interest and make a significant contribution.  The handling of the search needs to be revised as does the results and discussion.  I hope the comments below provide sufficient detail regarding the current problems.

I have made some comments below about the retrofitting of the later search that has identified 6 additional articles.  From what I can see, inclusion of these final 6 articles has not undergone the required screening processes.  To be honest, it does look as though the manuscript was prepared in 2019 and the authors have decided to update but not go through the same procedures as followed for the articles in the original search.  It makes no sense to the reader to split the searches in the way that it has been done.  The article needs to be revised and based on the search conducted in 2022, not the search conducted in 2019 with an add-on.

Section 2.2 mentions the search including dates up to 2019 and Table 1 gives the last date of the search as November 2019.  Figure 1 indicates another search was conducted in January 2022.  I assume it is Section 2.2 and Table 1 that need correction to include the new search in January 2022.

It doesn't appear as though the data extraction or quality analysis have been conducted for the 6 papers included in the new search (Tables 2 & 3) or in the overall analysis.

The PRISMA Flowchart (Figure 1) is of poor quality and difficult to read.  It also needs a title.

It would be useful to have more information about the decision to exclude studies with a focus on neurodiverse participants.  Did any of the studies included have neurodiverse participants even though this wasn't the focus of the study? 

Table 3 has author citations in the far right column covering data analysis framework.  This should be changed to a type of analysis rather than an author citation.

PsychInfo = PsycInfo

Figure 2 is difficult to read.  I am also not sure if it should be included as it doesn't seem to give any weighting to the quality of the study, the number of studies with the sub/theme etc.  It potentially distorts the findings.

The statement in the discussion "Enjoyment was discussed in the majority of studies (N=13)" seems to come from the original search with 20 studies and not include the last 6 studies.  The same is true for Table 4.

I didn't see the need for illustrative quotes and it consumed a lot of the discussion.  Readers should go to the original studies for this level of data.  The focus needs to be much more on the themes identified and a deeper analysis of the themes would be welcome e.g. connecting to the 'Background' section.

Author Response

This review is likely to be of interest and make a significant contribution.  The handling of the search needs to be revised as does the results and discussion.  I hope the comments below provide sufficient detail regarding the current problems.

Thank you to the reviewer for their detailed comments on the manuscript. We have addressed all of the comments below and the changes can be viewed in the manuscript. There does appear to have been an error in the original submission upload where the tables in particular were not up to date (I thought I had overwritten this file, but evidently this was not successful). We have therefore made substantial changes to certain sections of the manuscript, particularly the tables.

I have made some comments below about the retrofitting of the later search that has identified 6 additional articles.  From what I can see, inclusion of these final 6 articles has not undergone the required screening processes.  To be honest, it does look as though the manuscript was prepared in 2019 and the authors have decided to update but not go through the same procedures as followed for the articles in the original search.  It makes no sense to the reader to split the searches in the way that it has been done.  The article needs to be revised and based on the search conducted in 2022, not the search conducted in 2019 with an add-on.

The wording of section 2.2 has been amended to clarify that the same procedure was followed in the updated search as the original one. The original search did take place in 2019 and, following Cochrane guidance, the review was updated before submission in early 2022 to ensure that the review included the most up to date evidence. The manuscript has been updated to now reflect the new evidence and all 26 articles.

Section 2.2 mentions the search including dates up to 2019 and Table 1 gives the last date of the search as November 2019.  Figure 1 indicates another search was conducted in January 2022.  I assume it is Section 2.2 and Table 1 that need correction to include the new search in January 2022.

Table 1 has been updated with the correct date for the final database search- this was a typographical error in the previous submission.  

It doesn't appear as though the data extraction or quality analysis have been conducted for the 6 papers included in the new search (Tables 2 & 3) or in the overall analysis.

Data extraction and quality analysis were conducted on all papers, including the 6 papers identified through the updated search. Tables 2 and 3 now contain details of the quality appraisal and data extraction for all 26 papers. We hope that the amendments to the main text based on other comments also make this clearer.  

The PRISMA Flowchart (Figure 1) is of poor quality and difficult to read.  It also needs a title.

The figure has been updated in line with other comments and has been made higher resolution to improve quality. As per journal requirements a figure caption is presented underneath, but we have added in the stages of searching according to PRISMA.

It would be useful to have more information about the decision to exclude studies with a focus on neurodiverse participants.  Did any of the studies included have neurodiverse participants even though this wasn't the focus of the study? 

Whilst we did not included studies that actively recruited participants on the basis of neurodiversity we acknowledge that some of the studies may have included neurodiverse participants (and indeed participants with specific medical diagnoses etc). However, none of the authors specify any neurodiversity within the sample or identify quotations on this basis, therefore we can not provide clarity on this. Accordingly, we have amended the exclusion criteria to acknowledge that studies may included participants from these groups (section 2.2).

Table 3 has author citations in the far right column covering data analysis framework.  This should be changed to a type of analysis rather than an author citation.

All citations have been removed from this column so it now just contains the analysis framework.

PsychInfo = PsycInfo

Thank you for pointing out this error. This has been amended throughout the manuscript

Figure 2 is difficult to read.  I am also not sure if it should be included as it doesn't seem to give any weighting to the quality of the study, the number of studies with the sub/theme etc.  It potentially distorts the findings.

We have removed figure 2 from the manuscript.

The statement in the discussion "Enjoyment was discussed in the majority of studies (N=13)" seems to come from the original search with 20 studies and not include the last 6 studies.  The same is true for Table 4.

This was a typographical error and we have amended the text to read ‘Enjoyment was discussed in half of the studies (N=13)’. We have checked all of these descriptions within the main text and the numbers and proportions within tables 4-6 to ensure that they reflect the 26 studies. It should also be noted that tables 4-6 now contain example quotations from across the 26 studies.

I didn't see the need for illustrative quotes and it consumed a lot of the discussion.  Readers should go to the original studies for this level of data.  The focus needs to be much more on the themes identified and a deeper analysis of the themes would be welcome e.g. connecting to the 'Background' section.

Noted. We have removed the example illustrative quotations from the main text and readers can still find these examples in the tables. We have presented a separate results and discussion section. The results provide a descriptive account of the current evidence base (where we included an example illustrative quotation in the original submission) and the discussion provides a more in depth interpretation of this evidence and places it in the wider context, such as that presented in the background section. We feel this is a logical way of structuring the manuscript and so have maintained these distinct sections. However, in response to this comment we have revised the discussion to ensure that each section better reflects and relates to the information presented in the background section.

Reviewer 2 Report

The purpose and rationale are clearly stated and the manuscript adds important  data and results  to the literature base. I have a few minor suggestions below. 

Abstract: UK and CASP are both used as acronyms without being spelled out first. 

Section 2.1 - Paragraph starting: "results were exported". The inclusion and exclusion criteria are included but they are not clearly stated. Please consider being more clear, creating a list, etc. with both inclusion and exclusion criteria. Also, please be more specific on why process data was excluded? Were their interview protocols examined to see what type of questions were asked? 

Section 2.2. Thank you for using CASP.  However, it was confusing to me to read about the articles included and screened using CASP prior to understanding how many articles were included through the review process. I would suggest only stating methods here and then discussing results of CASP in results. I recognize this may be a preference or journal request though. 

I would also like more discussion of why all articles were included even if there were  discrepancies with quality. Did you have a cut off point that  if an article did not meet x amount of criteria it would be excluded? If not, what was the purpose of CASP if all included? 

Author Response

The purpose and rationale are clearly stated and the manuscript adds important data and results  to the literature base. I have a few minor suggestions below. 

Thank you to the reviewer for their kind comments regarding the manuscript. We outline how each of their comments has been addressed below.

Abstract: UK and CASP are both used as acronyms without being spelled out first. 

The Abstract has been updated to include these acronyms written out in full.

Section 2.1 - Paragraph starting: "results were exported". The inclusion and exclusion criteria are included but they are not clearly stated. Please consider being more clear, creating a list, etc. with both inclusion and exclusion criteria. Also, please be more specific on why process data was excluded? Were their interview protocols examined to see what type of questions were asked? 

This paragraph has been amended to state the inclusion criteria more explicitly using bullet points. We have also amended the final sentence in this paragraph to be more specific about why process evaluation papers were excluded. This was largely due to the questions asked in these evaluations being very specific to the interventions rather then experiences of physical activity more broadly. We do think a synthesis of process evaluation’s would also provide interesting findings but would have a different aim of the present paper. We have left the reasons for exclusion in continuous prose as we feel it is important to justify these decisions, however we have reworded the section for clarity.  

Section 2.2. Thank you for using CASP.  However, it was confusing to me to read about the articles included and screened using CASP prior to understanding how many articles were included through the review process. I would suggest only stating methods here and then discussing results of CASP in results. I recognize this may be a preference or journal request though. 

This section (we think the reviewer meant section 2.3 but forgive us if we have misunderstood the comment) has been amended to include only the details of what the CASP tool is and how it was used in the present study. Details of the results of this quality appraisal are now included in the results (section 3.1).

I would also like more discussion of why all articles were included even if there were discrepancies with quality. Did you have a cut off point that  if an article did not meet x amount of criteria it would be excluded? If not, what was the purpose of CASP if all included? 

The purpose of assessing the quality appraisal tool has now been added to section 2.3. We did not approach the quality appraisal with a pre-determined criteria for inclusion or exclusion, however the key factor influencing the decision to include all studies is that they were all deemed to make a valuable contribution to the evidence base (the results have been amended to include a statement emphasising this- section 3.1). Therefore, the purpose of using the CASP tool was more to identify aspects of study methodology that have not been consistently done well, and therefore to suggest opportunities for the quality of research to improve in future. This has now been highlighted in the text.